# Integrating Machine Learning and Bulk and Single-Cell RNA Sequencing to Decipher Diverse Cell Death Patterns for Predicting the Prognosis of Neoadjuvant Chemotherapy in Breast Cancer

**DOI:** 10.3390/ijms26083682

**Published:** 2025-04-13

**Authors:** Lingyan Xiang, Jiajun Yang, Jie Rao, Aolong Ma, Chen Liu, Yuqi Zhang, Aoling Huang, Ting Xie, Haochen Xue, Zhengzhuo Chen, Jingping Yuan, Honglin Yan

**Affiliations:** Department of Pathology, Renmin Hospital of Wuhan University, Wuhan 430060, China; xianglingyan@whu.edu.cn (L.X.); 2020305233042@whu.edu.cn (J.Y.); rm002228@whu.edu.cn (J.R.); 2024283020213@whu.edu.cn (A.M.); 15809905150@163.com (C.L.); 2021305231104@whu.edu.cn (Y.Z.); huangaoling@whu.edu.cn (A.H.); xietingting@whu.edu.cn (T.X.); 2024283020214@whu.edu.cn (H.X.); chenzhengzhuo@whu.edu.cn (Z.C.); yuanjingping@whu.edu.cn (J.Y.)

**Keywords:** neoadjuvant chemotherapy, breast cancer, machine learning, programmed cell death, bulk and single-cell RNA sequencing

## Abstract

Breast cancer (BRCA) continues to pose a serious risk to women’s health worldwide. Neoadjuvant chemotherapy (NAC) is a critical treatment strategy. Nevertheless, the heterogeneity in treatment outcomes necessitates the identification of reliable biomarkers and prognostic models. Programmed cell death (PCD) pathways serve as a critical factor in tumor development and treatment response. However, the relationship between the diverse patterns of PCD and NAC in BRCA remains unclear. We integrated machine learning and multiple bioinformatics tools to explore the association between 19 PCD patterns and the prognosis of NAC within a cohort of 921 BRCA patients treated with NAC from seven multicenter cohorts. A prognostic risk model based on PCD-related genes (PRGs) was constructed and evaluated using a combination of 117 machine learning algorithms. Immune infiltration analysis, mutation analysis, pharmacological analysis, and single-cell RNA sequencing (scRNA-seq) were conducted to explore the genomic profile and clinical significance of these model genes in BRCA. Immunohistochemistry (IHC) was employed to validate the expression of select model genes (*UGCG*, *BTG22*, *TNFRSF21,* and *MYB*) in BRCA tissues. We constructed a PRGs prognostic risk model by using a signature comprising 20 PCD-related DEGs to forecast the clinical outcomes of NAC in BRCA patients. The prognostic model demonstrated excellent predictive accuracy, with a high concordance index (C-index) of 0.772, and was validated across multiple independent datasets. Our results demonstrated a strong association between the developed model and the survival prognosis, clinical pathological features, immune infiltration, tumor microenvironment (TME), gene mutations, and drug sensitivity of NAC for BRCA patients. Moreover, IHC studies further demonstrated that the expression of certain model genes in BRCA tissues was significantly associated with the efficacy of NAC and emerged as an autonomous predictor of outcomes influencing the outcome of patients. We are the first to integrate machine learning and bulk and scRNA-seq to decode various cell death mechanisms for the prognosis of NAC in BRCA. The developed unique prognostic model, based on PRGs, provides a novel and comprehensive strategy for predicting the NAC outcomes of BRCA patients. This model not only aids in understanding the mechanisms underlying NAC efficacy but also offers insights into personalized treatment strategies, potentially improving patient outcomes.

## 1. Introduction

Breast cancer (BRCA) is a high-incidence malignancy that threatens women’s health. In recent years, neoadjuvant chemotherapy (NAC) performed before chemoradiotherapy and surgery, has become an important part of the standard treatment for BRCA, especially for patients with human epidermal growth factor receptor-2 (HER2)-positive and triple-negative breast cancer (TNBC) [1]. Current studies have revealed that patients can achieve tumor downstaging through NAC, which makes inoperable patients eligible for conservative surgery. In particular, it can help achieve a pathological complete response (pCR) and promote better long-term results in patients [2]. However, due to the heterogeneity of patients’ responses to chemotherapy, even the same molecular subtype of BRCA patients still have great differences in the treatment outcome of NAC [3]. Hence, there is a pressing need to explore novel biomarkers and construct more accurate prognostic prediction models to anticipate treatment efficacy and prognosis of BRCA patients to NAC [4].

Programmed cell death (PCD), which is prevalent in all organisms, is an active and orderly way of gene-regulated cell death. When cells are exposed to internal and external environmental stimuli, suicide protective measures initiated by the regulation of related genes can induce the activation of many downstream pathways, resulting in the eventual rupture and death of cells. Several patterns of death have been well studied in PCD, including apoptosis, pyroptosis, anoikis, PANoptosis, necroptosis, ferroptosis, cuproptosis, autophagy, parthanatos, lysosome-dependent cell death, alkaliptosis, oxeiptosis, NETosis, netotic cell death, immunogenic cell death, paraptosis, methuosis, entosis, entotic cell death, etc. [5,6].

In NAC for BRCA, numerous anticancer drugs have been shown to limit tumor progression by inducing PCD. For instance, several chemotherapy agents, such as doxorubicin, paclitaxel, and cisplatin, can induce pyroptosis in BRCA cells. This is achieved by activating the caspase-dependent ROS/JNK signaling pathway. Additionally, downstream pathways regulated by the gasdermin family of proteins are involved. Together, these mechanisms contribute to the anti-tumor effects of these treatments [7,8,9]. In addition, a study has shown that the synergistic effect of pyrotinib and chrysin enhances autophagy in HER2-positive BRCA via a previously unrecognized miR-16-5p/zinc finger and BTB/POZ domain-containing family protein 16 (ZBTB16)/glucose-6-phosphate dehydrogenase (G6PD) axis [10]. Not coincidentally, another study has shown that a treatment combining 2 μM auranofin (AF) with the proteasome inhibitor bortezomib (Bz), or TrxR1 knockdown in combination with Bz, selectively triggered paraptosis in BRCA cells, thereby offering potential therapeutic benefits for BRCA treatment [11]. The above studies suggest that the anticancer mechanism of certain classical chemotherapy drugs may be highly correlated with PCD, which may help to offer innovative therapeutic approaches for BRCA management. However, these studies have only focused on a single pattern of PCD, and the extensive relationship between multiple forms of PCD and NAC for BRCA is still unclear.

In this study, we focused on 19 PCD patterns and successfully identified 20 PCD-related differentially expressed genes (DEGs) associated with the prognosis of NAC for BRCA. A total of 10 machine learning algorithms for dimensionality reduction and regression were gathered and subsequently merged into 117 comprehensive machine learning algorithms. Based on this, we constructed a PCD-related gene (PRG) prognostic risk model capable of forecasting the outcomes of NAC in BRCA patients. We incorporated immune infiltration analysis, mutation analysis, pharmacological analysis, and single-cell RNA sequencing (scRNA-seq) to examine the genetic characteristics and clinical implications of these model genes in BRCA. Furthermore, we also validated the expression of PCD-related genes in clinical samples from BRCA patients who received NAC. This unique model offers a novel and comprehensive strategy for precisely predicting the prognosis of BRCA patients receiving NAC and evaluating their drug sensitivity, thus promoting a more personalized and effective treatment regimen and ultimately enhancing the clinical prognosis of patients.

## 2. Results

### 2.1. The Workflow of This Study

To develop and evaluate our predictive model, we performed an extensive reanalysis of several publicly available datasets comprising BRCA patients treated with NAC. These cohorts included GSE10886 [12], GSE16446 [13], GSE20685 [14], GSE22226 [15], GSE25055 [16], GSE25065 [16], and GSE32603 [17]. We screened the genes associated with 19 patterns of PCD, including apoptosis, pyroptosis, anoikis, PANoptosis, necroptosis, ferroptosis, cuproptosis, autophagy, parthanatos, lysosome-dependent cell death, alkaliptosis, oxeiptosis, NETosis, netotic cell death, ICD, paraptosis, methuosis, entosis, and entotic cell death. After sorting and deduplicating, a total of 2091 PCD-related genes (PRGs) were finally obtained. The workflow of the study is illustrated in Figure 1.

### 2.2. Identification of NAC-Related PRG Clusters

The datasets were first subjected to quality assessment, background correction, normalization, and batch effect removal (Figure 2A,B). Seven independent datasets were divided randomly into training and validation subsets in an 8:2 ratio. To deeply analyze the potential impact of the PRG pathways on NAC for BRCA, we extracted an expression matrix of 2091 PRGs from the meta cohort. Through differential analysis (all adjusted *p* < 0.05 and |log2FC| >  0.5), 923 DEGs were identified. We then performed a one-way Cox regression analysis of the survival information of PRGs in seven independent training datasets and the meta training cohort. The PRGs that showed a significant correlation with survival in five or more datasets were identified as the NAC-related PRGs in BRCA (Figure 2C). A total of 10 PRGs were finally obtained. By applying K-means clustering, we classified BRCA samples of a meta cohort based on the expression profiles of 10 PRGs and determined the optimal K value (k = 3) (Figure 2D). As depicted in Figure 2E, the entire cohort of BRCA patients was divided into three PRG clusters. We performed a differential analysis (adjusted *p* < 0.05 and |log2FC| >  0.5) among three PRG clusters, and finally obtained 31 DEGs, which were regarded as PCD-related DEGs. The expression correlations of the 31 PCD-related DEGs are shown in Figure 2F. To investigate the presence of populations with differential expressions of PRGs, we performed unsupervised clustering using 10 PRGs across eight independent cohorts. A Kaplan–Meier analysis was then performed among different clustering groups; the results showed significant survival differences in all seven independent datasets as well as in the meta cohort (Figure 2G–N).

### 2.3. Variant Landscape of PCD-Related DEGs in BRCA Patients

We analyzed in depth the potential association between PRG clusters and clinical characteristics. Figure 3A demonstrates the differential distribution of the 31 PCD-related DEGs and the six types of clinical characteristics among different PRG clusters. Both age and tumor grade showed significant distributional differences among different PRG clusters. All of the 31 PCD-related DEGs showed significant expression differences among different PRG clusters. GO and KEGG enrichment analyses demonstrated that many of the PCD-related DEGs were involved in immune processes related to T cells (Figure 3B,C).

### 2.4. Immune Infiltration Characteristics in the Distinct PRG Clusters

We analyzed the differences in TME components among PRG clusters by eight immune infiltration analysis tools in the meta cohort (Figure 4A). Differences in the expression of immune-related cytokines and their receptors were also analyzed to indirectly reflect the differences in the immune levels of PRG clusters (Figure 4B). The results demonstrate that notable disparities were observed in the levels of multiple immune cells, and remarkable variations were also observed in the expression of multiple immune-related cytokines, immune checkpoints, and their receptors among the 3 PRG clusters.

### 2.5. Construction and Evaluation of the PRGs Prognostic Risk Model via the Machine Learning-Based Integrative Procedure for BRCA Patients

To attain a high accuracy and scientific validity of the prognostic gene signature for BRCA patients who received NAC, we integrated the previously obtained 10 NAC-related PRGs with the 31 PCD-related DEGs screened from the above-mentioned 3 PRG clusters. Finally, 35 PRGs were obtained after deduplication for subsequent model construction. Based on the 35 PRGs, 10 distinct machine learning algorithms were employed, such as the Random Forest algorithm, LASSO regression, GBM algorithm, SVM algorithm, etc., and subsequently, these algorithms were combined to form 117 comprehensive machine learning algorithms for model construction.

The model training was performed using the meta cohort training set. In order to obtain a more accurate prognostic prediction ability on the validation set, the optimal model was taken by using random seeds and parameter tuning of the RSF algorithm for multiple training. Notably, the training results showed that the model combinations of CoxBoost + Survival-SVM exhibited excellent prediction ability, with the highest average C-index being 0.772 (Figure 5A). These two algorithms were subsequently used for gene selection and model development. The model retained a total of 20 PRGs (*ESR1*, *XBP1*, *MYB*, *VAV3*, *UGCG*, *CCNG2*, *BTG2*, *FBXL5*, *ASNS*, *NINJ1*, *IGF1R*, *RSU1*, *TNFRSF21*, *RARRES3*, *NFKBIN*, *IFNGR2*, *ABL2*, *ACVR1*, *LAMA3*, *VDAC2*) by dimensionality reduction and gave them different weight coefficients. We then categorized the patients into high-risk and low-risk subgroups using the median score of the cohort. The Kaplan–Meier survival curve revealed significant survival differences between the high- and low-risk subgroups in seven independent and meta cohort validation sets (Figure 5B–I). The ROC curves for 1 to 3 years in seven independent and meta cohort validation sets also showed excellent predictive ability of the model (Figure 5J–L).

Then, we further verified the predictive effect of this model on the survival prognosis of BRCA following NAC. Multivariate Cox regression analysis revealed that this model had a strong association with the survival of BRCA patients who received NAC and was independent of other clinical factors (Figure 6A). After that, we also selected six similar BRCA NAC prognostic models to compare their prediction levels with that of our model. As shown in Figure 6B–J, our model performed excellently in most of the datasets and significantly outperformed the other prognostic models. When comparing our model with other clinical factors, our model still demonstrated a high level of prediction in most of the datasets (Figure 6K–R). This further confirmed the correctness and scientific validity of our choices made through cluster analysis, differential analysis, and modeling methods.

### 2.6. Clinicopathologic Features and TIME Analysis Based on the PRGs Prognostic Risk Model

In order to more clearly determine the differences between the high- and low-risk subgroups as well as the associations between the prognostic model and clinicopathologic features in BRCA patients, we analyzed the differences in the levels of 20 model genes and 10 clinical factors between the high- and low-risk subgroups, as shown in Figure 7A. The analysis revealed that most model genes exhibited markedly distinct expression patterns when comparing high- and low-risk patient subgroups. Notably, several clinical pathological parameters, including age, HER2, and RCB grading, also exhibited markedly distinct variations across risk-stratified groups, which suggested that these clinicopathologic features may be potentially associated with the level of risk scores.

We subsequently assessed the percentage of immune cells within the datasets via CIBERSORT. As shown in Figure 7B, we found that macrophages as well as T-cells were more predominant in the datasets, whereas B-cells and NK-cells were underrepresented. The ESTIMATE computational method was additionally employed to calculate stromal and immune component indices, ESTIMATE-based evaluation metrics, and neoplastic cellularity in BRCA cases. It revealed that the stromal score and the immune score showed notable differences between the high- and low-risk score subgroups, displaying lower stromal scores and higher immune scores in high-risk PRGs score subgroups (Figure 7C–F). Afterwards, we analyzed the TIME components and immune-related cytokines in the high- and low-risk subgroups using various immune infiltration analysis tools. The results demonstrated that there was a more significant difference in the levels of multiple immune cells and immune-associated cytokines between the two subgroups, which suggested distinct immune infiltration characteristics between the high- and low-risk score subgroups (Figure 8). Thus, this finding suggested a potential correlation between the PRG risk scores and the immune microenvironment. It is speculated that the genes incorporated within these models might exert a certain influence on the formation of the immune microenvironment.

### 2.7. Single-Cell Transcriptome Analysis Demonstrates the Influence of Model Genes on the TME

In the gene chip transcriptomes, the analysis of the transcriptional levels involves analyzing tumor cells and various types of cells in the microenvironment all together. The results obtained from such analysis are inevitably biased, and it is impossible to determine which cells the differences in gene expression originate from. Therefore, we validated the previous conclusions by using a single-cell RNA sequencing database from BRCA (EMTAB8107). As shown in Figure 9A, we first analyzed the single-cell dataset by UMAP dimensionality reduction clustering and cell labeling to identify a total of 11 cell populations. Once effective cell clustering is achieved, expression levels of various model genes across different cell populations are assessed. Our analysis revealed that most model genes were highly expressed in plasma cells, tumor cells, monocytes, macrophages, etc. Figure 9B–H illustrates the distribution of some model genes among different cell populations. Such results validated the previous speculation that model genes are differentially expressed in the immune microenvironment of cancer tissues and have a significant effect on the formation of the immune microenvironment.

### 2.8. Mutation Abundance Analysis

Genes that are crucial for the prognosis of BRCA patients receiving NAC may have higher mutation levels, which in turn, indirectly contribute to the differences in immune infiltration as well as prognostic outcome. Thus, we analyzed the model genes by placing them in the BRCA mutation database from TCGA. As shown in Figure 10A–G, nonsense mutations accounted for the largest proportion of mutations, and among the SNPs, alterations in cytosine occur most frequently. Mutations in model genes such as *LAMA3* and *IGF1R* are predominant, which also confirms the important potential roles of these genes in NAC for BRCA.

As for CNV, we analyzed the frequency of CNV mutations in 20 model genes in 31 type of cancers by pan-cancer analysis. Among numerous cancers, regardless of whether it is amplification or reduction, the mutation frequency of BRCA was relatively high. This is particularly evident in model genes like *ABL2* and *BTG2*, as shown in Figure 10H. Such abundant varieties and high frequencies of mutations imply that the heterogeneity of BRCA patients is extremely complex. Therefore, we conclude that these model genes may play a significant role in the prognosis of NAC for BRCA.

### 2.9. Validation of Immunohistochemical Staining of Model Genes from HPA Database

In order to verify the real existence of the above model genes in BRCA patients, we validated the immunohistochemical staining of model genes from the HPA database. As shown in Figure 11, many of the model genes were expressed in BRCA tissues. This verifies the existence and important role of model genes in BRCA.

### 2.10. The Prognostic Model Is Associated with the Response to NAC in BRCA

Due to the fact that patients have significant differences in their sensitivity to different NAC drugs, this also leads to heterogeneity in the efficacy of BRCA patients following NAC. To investigate whether our prognostic model is associated with different drug sensitivities, we conducted an association analysis between NAC drugs and genes within the model through the GDSC and CTRP pharmacology databases. As shown in Figure 12A,B, the discoveries supported that several model genes were highly tied to the sensitivities of multiple chemotherapy drugs. This implied that the prognostic model has potential value for use in predicting drug sensitivities.

The RCB grading is a comprehensive system for assessing the response to NAC in BRCA. It classifies patients into RCB classes from RCB-0 to RCB-III. A lower RCB grade means a better response to chemotherapy, and a higher grade indicates a poorer one. To further validate the predictive capacity of model gene expression in clinical specimens regarding the efficacy of NAC, we selected four genes (*UGCG*, *TNFRSF21*, *MYB*, *BTG*) that exhibited a remarkable correlation with the sensitivity to multiple chemotherapy drugs in the pharmacological analysis for in-depth validation. We conducted a retrospective collection of the pathological specimens from 56 BRCA patients who had previously received NAC. The postoperative pathological specimens of these patients had already undergone RCB grading assessment. Subsequently, we carefully retrieved the core needle biopsy pathological specimens of the same patients prior to NAC for IHC staining (Figure 12C).

Due to the limited number of patients with postoperative RCB-I in our sample and considering that both patients with postoperative RCB-0 and RCB-I exhibited a favorable response to NAC, we combined RCB-0 and RCB-I into one group for analysis when performing the chi-square test and Fisher’s exact test. The chi-square test and Fisher’s exact test indicated that postoperative RCB classification demonstrated a statistically significant correlation with the expression levels of UGCG, BTG2, TNFRSF21, and MYB, as well as lymph node metastasis in BRCA patients prior to NAC (Table 1). According to the clinical significance of RCB, a lower RCB classification indicates better efficacy of NAC. Therefore, we consider that patients with postoperative RCB-0 and RCB-I have good efficacy of NAC, while patients with RCB-II and RCB-III have poor efficacy of NAC. A subsequent investigation of associations demonstrated that UGCG and TNFRSF21 expression levels showed a positive relationship with NAC, whereas the expression of BTG2 and MYB, as well as the lymph node metastasis, were negatively correlated with the efficacy of NAC (Table 2). Specifically, patients were more likely to achieve a higher RCB classification when they were negative for UGCG and TNFRSF21, but positive for BTG2 and MYB, with no lymph node metastasis.

To evaluate potential predictors of NAC response, both single- and multiple-factor regression models were employed, incorporating key clinicopathological variables such as UGCG, BTG2, TNFRSF21, MYB, age, ER, PR, HER2, and Ki-67. The single-variable analysis revealed that high expressions of UGCG and TNFRSF21, as well as low expressions of BTG2 and MYB, were favorable factors associated with a better efficacy of NAC. The multivariate analysis further pointed out that the expression levels of UGCG, BTG2, TNFRSF21, and MYB were independent influencing factors for the efficacy of NAC (Appendix A).

## 3. Discussion

NAC has emerged as a pivotal therapeutic strategy in the management of BRCA. Various patterns of PCD have long been recognized as being closely linked to the development and metastasis of human tumors [24]. Despite its substantial importance, the intricate interplay between diverse forms of PCD and the prognosis of NAC in this context remains veiled in obscurity. To date, no prognostic model that comprehensively incorporates the various patterns of cell death has been developed to forecast the outcome of NAC for BRCA.

In this study, we focused on 19 PCD patterns and developed a signature comprising 20 PCD-related DEGs (*ESR1*, *XBP1*, *MYB*, *VAV3*, *UGCG*, *CCNG2*, *BTG2*, *FBXL5*, *ASNS*, *NINJ1*, *IGF1R*, *RSU1*, *TNFRSF21*, *RARRES3*, *NFKBIA*, *IFNGR2*, *ABL2*, *ACVR1*, *LAMA3*, *VDAC2*) by employing machine learning algorithms. *ESR1*, a gene encoding ER, commonly undergoes genetic alterations in hormone therapy-resistant ER-positive BRCA, promoting ligand-independent activation and metastasis [25]. *XBP1* acts as a key regulator in the unfolded protein response (UPR) through the regulation of proteostatic mechanisms within the endoplasmic reticulum, ensuring proper cellular stress management [26]. *MYB* family members contribute to tumor initiation and maintenance via their critical involvement in fundamental biological functions including proliferation, specialization, and viability maintenance, with aberrant expression linked to oncogenesis [27]. *VAV3* interferes with multiple biological mechanisms such as proliferation, invasion, migration, cell cycle control, epithelial–mesenchymal transition (EMT), and apoptosis, which in turn facilitates cancer progression [28]. *UGCG* serves as the pivotal catalytic component in glycosphingolipid biosynthesis and regulation, generating glucosylceramide (GlcCer) de novo, and its increased synthesis is linked to enhanced proliferation in various cancers, promoting pro-cancerous processes [29]. The cyclin family gene *CCNG2*, which plays a key inhibitory role, is critical in inhibiting tumorigenesis and malignant advancement [30]. *BTG2* belongs to the BTG/Tob family. It is essential for varied cellular activities, such as cell growth, proliferation, and apoptosis, as it regulates several essential biological processes, including transcription and translation [31]. *FBXL5* is a key E3 ubiquitin ligase that regulates iron homeostasis by mediating the degradation of iron-responsive element-binding proteins IRP2 [32]. *ASNS* is an important factor in regulating CD8+ T cell function. Its expression peaks during the effector phase of T cell differentiation and declines with the transition of T cells to the memory phase [33]. *NINJ1* is located on the cell surface. It contains two transmembrane regions and serves as a pivotal regulator in the induction of programmed membrane rupture (PMR) [34]. *IGF1R* is a transmembrane tyrosine kinase. It plays such a vital role in controlling cell growth and differentiation that its dysfunction can trigger malignant conversion and neoplastic progression [35]. *RSU1* is involved in cell-extracellular matrix (ECM) adhesion. It participates in BRCA metastasis by localizing focal adhesions through its interaction with PINCH1, serving as a connector to downstream signaling pathways [36,37]. *TNFRSF21* is part of the tumor necrosis factor/tumor necrosis factor receptor (TNF/TNFR) family and plays a vital role in controlling inflammation and immune responses. It also contributes to the degeneration of the mammary gland and helps defend against infections [38]. *RARRES3* is essential for controlling tumor cell adhesion and differentiation, with its downregulation linked to metastasis [39]. Yang et al. [40] explored how *NFKBIA* influences NF-κB signaling activity and inflammation associated with BRCA tumors, with their findings showing that *NFKBIA* functions as a suppressor of the NF-κB pathway. *IFNGR2*, along with *IFNGR1*, binds to IFN-γ and initiates the activation of the downstream JAK/STAT pathway to mediate its effects on target cells [41]. *ABL2* is engaged in the proliferation of mature T cells, cytokine production, and chemokine-directed T cell migration [42]. *ACVR1* gene encodes the TGF-β superfamily receptor ALK7 [43]. ALK7 functions primarily in cells and tissues with endocrine functions [44]. *LAMA3* is a widely studied methylated gene. It is implicated in the pathogenesis and prognosis of various malignancies [45]. *VDAC2* is a crucial component of the mitochondrial outer membrane, facilitating the transport of ATP/ADP, ions, NADH, and other metabolites. It is crucial in controlling apoptosis for it to interact with the pro-apoptotic protein BAK [46]. As can be seen from the above, these model genes play essential roles in tumorigenesis and progression, inflammation, the TME, and cell death. Some of these genes have been reported to be potentially associated with NAC efficacy in BRCA. For example, tumors with high ESR1 expression often have a lower pCR rate, possibly because such tumors rely on hormonal signaling rather than chemotherapy-sensitive pathways [47]. In addition, the IGF1R rs2016347 T allele and chemotherapy-induced IGF1R downregulation were jointly associated with improved NAC response [48]. However, the bond between some other PCD-related DEGs and NAC in BRCA is not yet understood.

Subsequently, we performed an integrative bioinformatic analysis to explore the genomic characteristics and therapeutic implications of these predictive genes in BRCA. The constructed PRGs prognostic model demonstrated excellent predictive accuracy, with a high C-index of 0.772, and was validated across multiple independent datasets. By comparing with other existing models, it is proven that this model has advantages in predicting BRCA patient prognosis after NAC, further highlighting its potential value in clinical practice.

We analyzed the differences across risk-stratified subgroups defined by the predictive model across various parameters. Our analysis revealed significant associations between the PRG risk signature and key clinical indicators including age, HER2 status, and RCB grading. RCB grading is a comprehensive system used in the context of NAC for BRCA [3,49]. It takes into account multiple parameters, including the size of the residual tumor, the number of involved lymph nodes, the cellularity of the primary tumor bed, and the presence of in situ carcinoma. These parameters are combined to calculate an RCB score, which classifies patients into different RCB classes, usually ranging from RCB-0 to RCB-III. The RCB grading system holds significant importance in predicting the efficacy of NAC in BRCA [50,51]. A lower RCB score, such as RCB-0, indicates a more favorable prognosis, while a higher RCB score (e.g., RCB-III) indicates a less favorable response to treatment, with a greater amount of residual cancer, which is potentially connected to a poorer prognosis and a higher risk of disease recurrence. Here, significant RCB grading differences between the high-risk and low-risk subgroups suggested that the model is related to the efficacy of NAC for BRCA.

Tumor cells have developed mechanisms to avoid immune detection. These mechanisms allow them to withstand the impact of therapeutic agents, ultimately promoting their survival and progression [52]. An analysis of the immune microenvironment demonstrated marked variations in immune cell infiltration and cytokine expression among different PCD-related gene clusters. Various patterns of PCDs have been stated to influence the recruitment and activation of immune cells by releasing damage-associated molecular patterns, thus regulating immune infiltration; meanwhile, the metabolites and signaling molecules generated during PCD alter the composition and state of the immune microenvironment, and in turn, the cytokines and immune cells in the immune microenvironment also affect the occurrence and progression of PCD [53,54,55]. This indicates that PCD pathways may engage with the immune system to influence tumor progression and treatment response. This study also identified notable variations in the immune infiltration characteristics between the high-risk and low-risk subgroups based on the PRGs prognostic model. It is noteworthy that the high-risk PRG score subgroups showed lower stromal scores and higher immune scores, suggesting a more active immune-mediated anti-tumor process in these cases. However, this could also imply a more complex TME that might be more challenging to treat. The analysis of TIME components and immune-related cytokines further validates the distinct immune infiltration characteristics between the two subgroups. These differences in immune cell levels and cytokines not only support the correlation between PRG risk scores and the immune microenvironment but also provide potential insights into the fundamental biological pathways.

In recent years, scRNA-seq has emerged as an indispensable tool in the in-depth exploration of TME [56,57,58]. By virtue of its unique capacity to conduct gene expression profiling analysis precisely at the single-cell level, scRNA-seq enables in-depth investigation and precise classification of heterogeneous cellular components within the TME. Notably, the present study capitalizes on scRNA-seq to uncover the alterations in cellular heterogeneity of model genes at the level of individual cells. The results vividly demonstrated that model genes exhibit high expression levels across multiple cell types, including plasma cells, tumor cells, monocytes, and macrophages. This finding further cements the crucial significance of these model genes within the TME, highlighting their potential as key players in regulating tumor-related biological processes and as promising targets for future therapeutic interventions.

Mutation abundance analysis of model genes is also performed. The high proportion of nonsense mutations and the frequent occurrence of cytosine alterations among SNPs suggest unique mutational patterns in these genes. The predominance of mutations in genes like *LAMA3* and *IGF1R* not only emphasizes their potential significance in NAC for BRCA but also implies that these mutations might be associated with the differences in immune infiltration and prognostic outcomes, as initially hypothesized. The relatively high mutation frequency of BRCA in CNV analysis across 31 types of cancers, especially in genes such as *ABL2* and *BTG2*, further underlines the extreme complexity of BRCA patient heterogeneity. These abundant mutations could potentially disrupt normal gene functions, leading to diverse responses to NAC. Overall, the complex mutational landscape of these model genes strongly indicates their crucial role in predicting the prognosis of NAC for BRCA patients.

Finally, this study provides insight into the relationship between model genes and drug sensitivity of NAC in BRCA patients. It is worth noting that several genes in the model are highly correlated with a variety of NAC drugs, suggesting that the model can be used as a predictor of NAC choice for BRCA patients and help to develop more effective treatment plans. The real existence of the above model genes in BRCA patients was validated from the HPA database. Moreover, complemented by IHC studies, we have unequivocally demonstrated that the expression levels of certain model genes in BRCA tissues are significantly associated with the RCB grading. As mentioned above, the RCB grading system plays a pivotal role in predicting the efficacy of NAC in BRCA patients. Specifically, the protein expression of UGCG and TNFRSF21 displayed a negative correlation with the RCB grading, which implied a positive correlation with the sensitivity to chemotherapy drugs. UGCG’s negative correlation with RCB may stem from GlcCer accumulation, which is the precursor of all glycosphingolipids (GSL) complexes. GSL is not only an important membrane component but also participates in apoptosis as a signaling molecule [59]. TNFRSF21 internalization in acidic microenvironment recruits caspase-8/GSDMC, inducing pyroptosis and correlating with lower RCB [60]. Conversely, the protein expression of MYB and BTG2 showed a positive correlation with the RCB grade, indicating a negative correlation with drug sensitivity. These findings are in perfect alignment with the results derived from pharmacological analysis. Through comprehensive multivariate analysis, we have established that these model genes remain independent prognostic factors that significantly influence the efficacy of NAC in patients. This discovery not only accentuates the critical role of model genes in accurately predicting the efficacy of NAC but also underscores the remarkable clinical value of the established prognostic model.

Our PRG-based prognostic model holds significant translational potential in clinical settings, particularly in guiding NAC treatment strategies for BRCA patients. Through the use of this model, clinicians could better stratify patients undergoing NAC, predict treatment response, and optimize therapeutic decisions accordingly. Patients identified as high-risk based on the PRG score may benefit from more intensive or alternative therapeutic regimens, such as incorporating immune checkpoint inhibitors or targeted therapies. Conversely, low-risk patients with a favorable PRG profile might achieve sufficient tumor regression with standard NAC regimens, potentially avoiding overtreatment and its associated toxicities. This stratification would facilitate a more tailored approach to patient management, improving outcomes while minimizing unnecessary treatment burdens. By combining this model with conventional clinical and pathological features, such as hormone receptor status, HER2 expression, and Ki-67 index, clinicians could make more informed decisions regarding NAC suitability and post-treatment management. Furthermore, as single-cell transcriptomics advances, using insights from scRNA-seq into the model could refine its predictive accuracy, helping to identify patients who may benefit from novel therapeutic combinations.

Although our study contributes meaningful knowledge into the clinical significance of the PRGs prognostic risk model in NAC for BRCA, certain limitations must be recognized. First, our analysis is based primarily on retrospective data, emphasizing the need for prospective studies to verify the clinical pertinence of our results. The complexity of cancer and its different histological subtypes require more thorough mechanistic and clinical studies to further investigate the impact of model scores in different cancer types. In addition, although we have validated the protein levels of some model genes by IHC, more levels of in vitro validation are lacking. This could provide further insight into their functional significance. The detection of PCD-related DEGs in this model is feasible using standard molecular techniques such as scRNA-seq, quantitative PCR (qPCR), or IHC. To enhance the clinical impact of this model, further prospective validation in larger, multi-center cohorts with long-term follow-up is necessary.

In conclusion, we are the first to integrate machine learning and bulk and scRNA-seq to investigate various cell death types for forecasting the prognosis of NAC in BRCA. The achievement of our study demonstrated a notable interconnection between the developed model and the survival prognosis, clinical pathological features, immune infiltration, TME, gene mutations, and drug sensitivity of NAC for BRCA patients. These results highlight the model’s promising utility in clinical settings for informing tailored therapeutic strategies.

## 4. Materials and Methods

### 4.1. Data Acquisition

We collected high-quality datasets on NAC for BRCA from the Gene Expression Omnibus (GEO) (https://www.ncbi.nlm.nih.gov/) and The Cancer Genome Atlas (TCGA) (https://portal.gdc.cancer.gov/). After individually screening these datasets, we obtained a total of seven high-quality clinical datasets and gene expression matrices of 921 BRCA patients treated with NAC, including GSE10886 [12], GSE16446 [13], GSE20685 [14], GSE22226 [15], GSE25055 [16], GSE25065 [16], and GSE32603 [17]. Batch effects between datasets were cleared by the sva R package. Seven independent datasets and the meta cohorts were randomly divided into training and validation sets with an 8:2 ratio. The datasets underwent quality assessment, background adjustment, normalization, and batch effect removal. scRNA-seq data were obtained from an EMTAB scRNA-seq dataset (EMTAB8107).

To create a more comprehensive collection of PCD pathway genes, we used the GeneCards database (https://www.genecards.org/) and the Gene Set Enrichment Analysis (GSEA) database (https://www.gsea-msigdb.org/gsea/msigdb/index.jsp, accessed on 20 November 2024) and identified genes from 19 PCD pathways. In total, 2091 PCD genes were gathered for further analysis.

### 4.2. Identification of NAC-Related PRGs

The first step involved filtering out PRGs according to the following differential expression criteria: all adjusted by *p* < 0.05 and |log2FC| >  0.5. We then performed a one-way Cox regression analysis of the survival information of PRGs in seven independent training datasets and the meta training cohort. The PRGs that showed a significant correlation with survival in five or more datasets were identified as the NAC-related PRGs in BRCA.

### 4.3. Consensus Unsupervised Clustering Analysis

We leveraged the “ConsensusClusterPlus” R package (version number 1.66.0) to conduct a consensus unsupervised clustering analysis of the PRGs through the utilization of the K-means algorithm. Agglomerative hierarchical clustering was conducted using a Spearman correlation-based distance measure, with 80% of the data points subjected to resampling across 10 iterations. The scree plot method was applied to ascertain the most suitable cluster count. The Kaplan–Meier method was used to compare the overall survival (OS) among the different clusters. We also investigated the potential affiliation between PRG clusters and clinical manifestations.

### 4.4. Functional Enrichment Analysis

The “clusterProfiler” R package (version number 4.10.1) was utilized to identify potential gene ontology (GO) and Kyoto Encyclopedia of Genes and Genomes (KEGG) pathways using the DEGs identified above, aiming to uncover the hidden biological activities and signaling pathways significantly bound up with these DEGs [61].

### 4.5. Tumor Immune Microenvironment (TIME) Analysis

The TIME components analysis was performed by a variety of algorithms, including Cell-type Identification by Estimating Relative Subpopulations of RNA Transcripts (CIBERSORT), Estimation of Proportion of Immune and Cancer cells (EPIC), immunophenoscore (IPS), Microenvironment Cell Populations-counter (MCPcounter), quanTIseq, single-sample Gene Set Enrichment Analysis (ssGSEA), Tumor Immune Estimation Resource (TIMER), and TME_signature. Furthermore, we utilized the “Estimation of Stromal and Immune cells in Malignant Tumor tissues using Expression data” (ESTIMATE) algorithm to calculate the stromal score, immune score, ESTIMA score, and tumor purity of BRCA patients.

### 4.6. Construction of the PRGs Prognostic Risk Model by Machine Learning

To achieve a robust consensus of PRGs with enhanced precision and reliability, we implemented an integrated methodology by combining 10 unique machine-learning algorithms and 117 unique algorithm configurations. This integration incorporated a diverse array of methodologies, including CoxBoost, survival support vector machine (survival-SVM), random survival forest (RSF), elasticnetwork (Enet), Lasso, stepwise Cox, Ridge, supervised principal components (SuperPC), partial least squares regression forex (plsRcox), and generalized boosted regression modeling (GBM).

The generation of the signatures followed a sequential process comprising the subsequent steps: (a) The PRGs identified earlier were analyzed using 117 distinct algorithm configurations to develop predictive models, employing leave-one-out cross-validation (LOOCV) within the experimental subset of the meta cohort. (b) To ensure robustness, all models underwent additional validation across 7 independent datasets (GSE10886, GSE16446, GSE20685, GSE22226, GSE25055, GSE25065, and GSE32603) and the validation set for the meta cohort. (c) The effectiveness of each model was assessed by calculating Harrell’s concordance index (C-index) across all validation datasets, with the model exhibiting the greatest average C-index identified as the most optimal. Then, we classified BRCA patients into high-risk and low-risk subgroups based on the median score of the group. Following this, we evaluated the prognostic significance by means of Kaplan–Meier curves. Moreover, Receiver Operating Characteristic (ROC) curves for 1 to 3 years in 7 independent and meta cohort validation sets were constructed to assess the prognostic efficacy of the model. Multivariate Cox regression analysis was performed to determine whether the model could serve as a standalone prognostic factor of the NAC prognosis for patients with BRCA. We also investigated the latent interaction between the PRG predictive risk model and clinical features.

### 4.7. Single-Cell Sequencing Analysis

The analysis of scRNA-seq data from the ArrayExpress database (https://www.ebi.ac.uk/biostudies/arrayexpress/studies/E-MTAB-8107, accessed on 20 November 2024) for BRCA was conducted using the “Seurat” (version number 5.2.1) and “SingleR” (version number 2.4.1) packages, adhering to a sequence of standardized quality control steps that included the “PercentageFeatureSet”, “SCTransform”, “RunPCA”, “FindNeighbors”, “FindClusters”, “RunUMAP”, and “FindAllMarkers” functions. Cell-type annotation was performed using the “SingleR” tool along with well-known markers referenced in the existing literature. Furthermore, the biological roles of marker genes within every cellular subtype were determined using the “ClusterGVis” (version number 0.1.2) and “org.Hs.eg.db” (version number 3.18.0) R packages [62].

### 4.8. Pharmacological Analysis

We used the GSCA website (https://guolab.wchscu.cn/GSCA/#/, accessed on 25 November 2024) for the pharmacological analysis of our data. We entered the model genes into the DRUG module of the GSCA website and selected the GDSC and CTRP databases for drug sensitivity analysis to obtain the final results of the correlation between the model genes and a variety of chemotherapeutic drugs.

### 4.9. Mutation Analysis

Based on the transcriptome data and clinical information related to BRCA from TCGA-BRCA, we performed an in-depth evaluation of gene mutations and copy number variations (CNVs) in genes linked to response to NAC. Mutation analysis was carried out employing the “maftools” R package (version number 2.18.0), identifying significant single-nucleotide polymorphisms (SNPs), insertions, deletions, and their frequencies. For CNV analysis across 31 cancer types, we used “CNTools” (version number1.58.0) in R studio to examine amplification and deletion patterns in 20 selected genes within the model. The mutation landscape of these genes was visualized in stacked bar charts, variant classification plots, and summary charts for SNP classes. The frequency of the top 10 mutated genes and CNVs across cancer types was displayed using custom scripts and ggplot2 (version number 3.5.1) to illustrate gene-specific alterations in BRCA, highlighting the complexity and heterogeneity in this cohort.

### 4.10. Validation of Immunohistochemical Staining from the HPA Database

To verify the presence and localization of model genes, pathology images were retrieved from the Human Protein Atlas (HPA) database (https://www.proteinatlas.org). Corresponding images were selected based on cancer type and gene. All data used were in accordance with the terms of use of the HPA database.

### 4.11. Clinical Sample Collection and Immunohistochemical Analysis of BRCA

Formalin-fixed, paraffin-embedded tissue samples were collected from 56 BRCA patients who had previously received NAC from July 2019 to February 2022 at the Renmin Hospital of Wuhan University for immunohistochemistry (IHC) analysis. The postoperative pathological specimens of these patients had already undergone a RCB grading assessment. The RCB score is computed as a non-discrete numerical value and then classified into four categories: RCB-0, which represents a pathologic complete response (pCR) with an RCB score of 0; RCB-I, where the RCB score satisfies 0 < RCB ≤ 1.36; RCB-II, with 1.36 < RCB ≤ 3.28; and RCB-III, when RCB > 3.28. Relevant clinical and pathological data were also collected.

We selected four genes (*BTG*, *TNFRSF21*, *UGCG*, and *MYB*) in the model that were closely related to the sensitivity to multiple chemotherapy drugs for immunohistochemical verification by using the core needle biopsy pathological specimens of the same patients prior to NAC. Tissue sections of 4 µm thickness, preserved in formalin and embedded in paraffin, were initially dewaxed using xylene and subsequently rehydrated through a sequential ethanol gradient. Endogenous peroxidase activity was suppressed by treating the sections using 3% H_2_O_2_ for 10 min at room temperature. Antigen retrieval was carried out by heating the sections in citrate buffer (10 mM, pH 6.0) in a microwave. The sections were then blocked with a 1:10 dilution of goat serum for 30 min at room temperature. Following this, the sections were incubated overnight at 4 °C with primary antibodies targeting BTG (rabbit, 1:100, Proteintech (Wuhan, China), 22339-1-AP), DR6 (E-4) (also named TNFRSF21, mouse, 1:100, Santa Cruz (CA, USA), sc-376873), UGCG (rabbit, 1:100, Proteintech (Wuhan, China), 12869-1-AP), and MYB (rabbit, ready-to-use, Tongling Biomedical (Xiamen, China), AR0841). Following a PBS wash, the sections were treated with HRP-linked secondary antibodies for 30 min at room temperature, followed by a 20 min incubation with broad-spectrum HRP-conjugated secondary antibodies. Nuclei were highlighted using hematoxylin counterstaining. Images were obtained with an Olympus optical microscope (Olympus Corporation, Tokyo, Japan).

The IHC score was determined by assessing both the staining intensity and the percentage of tumor cells exhibiting positive staining, as outlined below: IHC score  =  percentage score  ×  intensity score. The scoring criteria were established as follows: 0, <10%; 1, 10–25%; 2, 26–50%; 3, 51–75%; 4, >75%. The definition of the intensity scores was as follows: 1, light brown; 2, brown; 3, dark brown; 0, no staining. Four levels were designated for the expression: negative (IHC score  =  3), weak (IHC score >1 and ≤4), moderate (IHC score >4 and ≤8), and strong (IHC score > 9). The BRCA patients were divided into two groups according to their expression levels: UGCG-high (IHC score > 4) and UGCG-low (IHC score ≤ 4); BTG-high (IHC score > 4) and BTG-low (IHC score ≤ 4). Due to the overall low or absent expression of TNFRSF21 and MYB in the patient cohort, the grouping criteria used were different from those for UGCG and BTG2. Those with positive expressions in BRCA tissue are identified as the positive group, and those with negative expressions are identified as the negative group.

### 4.12. Statistical Analysis

Data processing and analysis were performed using R version 4.4.0 (2024-04-24), SPSS 27.0, along with Zstats 1.0 (www.zstats.net). Continuous variables were expressed as mean ± standard error and analyzed using either Student's t test or Wilcoxon rank sum test. Kaplan–Meier survival analysis was performed to calculate survival curves, and the log-rank test was employed to evaluate differences between groups. Categorical data were assessed using the chi-square test. The association between RCB classification and clinicopathological parameters was evaluated using the chi-square test. When over 20% of the cells in the contingency table had an expected frequency below 5, Fisher’s exact test was applied to ensure accuracy, as the chi-square test approximation may be unreliable under such conditions. Spearman correlation analysis was applied to examine the relationship between the efficacy of NAC and clinicopathological features. Additionally, logistic regression analysis was conducted to identify factors influencing the efficacy of NAC. A *p*-value of less than 0.05 was considered statistically significant.

## Figures and Tables

**Figure 1 ijms-26-03682-f001:**
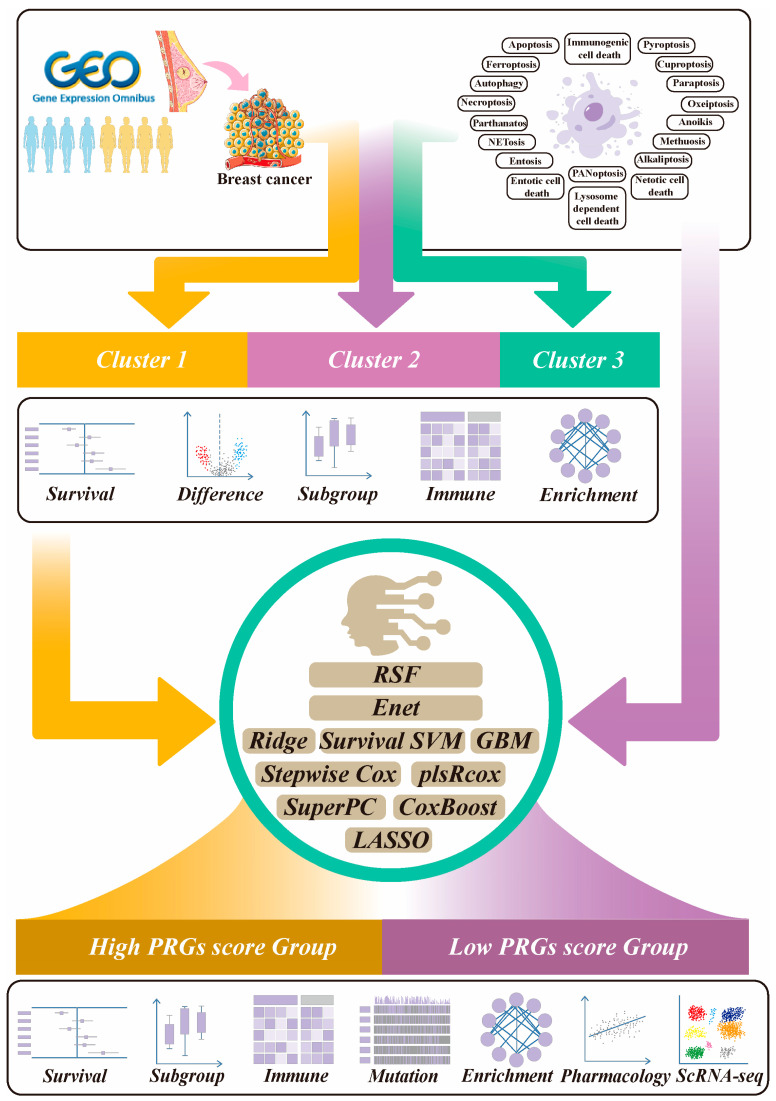
Graphic abstract of this study. Abbreviations: RSF, random survival forest; Enet, elastic network; Survival SVM, survival support vector machine; GBM, generalized boosted regression modeling; plsRcox, partial least squares regression forex; SuperPC, supervised principal components; PRGs, programmed cell death-related genes; ScRNA-seq, single-cell RNA sequencing.

**Figure 2 ijms-26-03682-f002:**
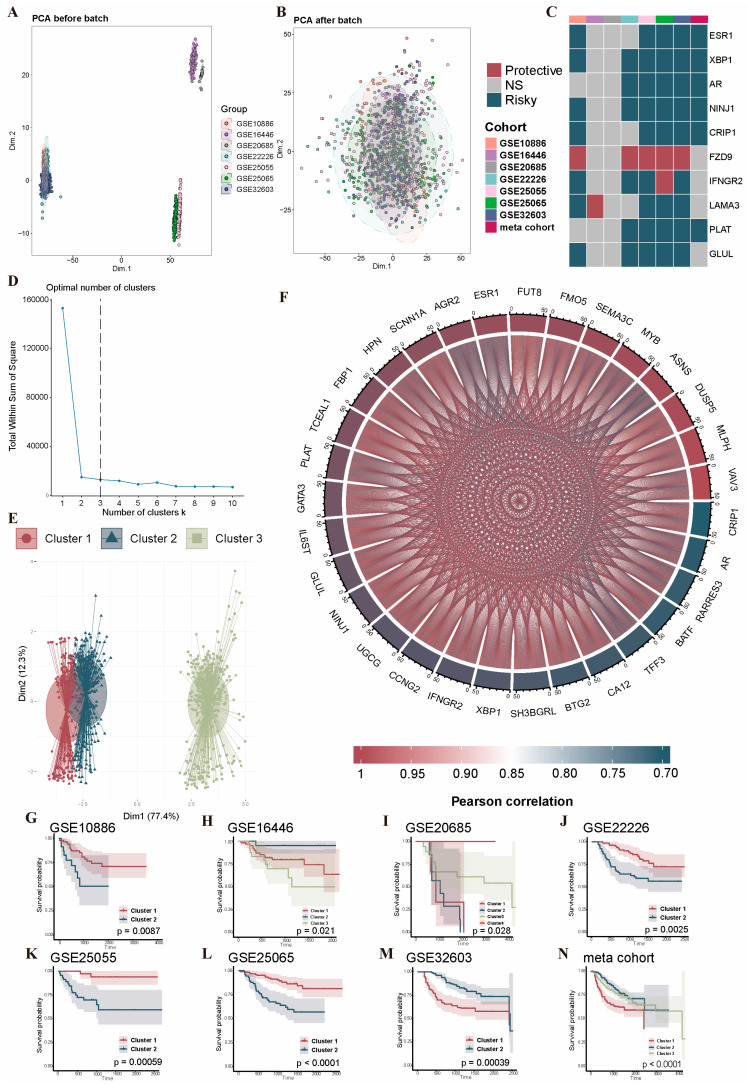
The identification of prognostics PRGs and the classification of NAC-related PRG clusters in BRCA. (**A**) The distribution of the 7 independent datasets in dimensions 1 and 2 before removal of batch effects. (**B**) The distribution of the 7 independent datasets in dimensions 1 and 2 after the removal of batch effects. (**C**) The prognostic value of the 10 survival-associated PRGs in the 7 datasets. (**D**) The scree plot of K-means clustering based on the meta cohort demonstrates that the optimal number of clusters is 3. (**E**) The meta cohort of BRCA patients was divided into 3 PRG clusters. (**F**) The interaction analysis among 31 PCD-related DEGs. (**G**–**N**) Kaplan–Meier survival analysis of PRG clusters for the 7 datasets and the meta cohort. Abbreviations: PRGs, programmed cell death-related genes; NAC, neoadjuvant chemotherapy; BRCA: breast cancer; PCD-related DEGs: programmed cell death-related differentially expressed genes.

**Figure 3 ijms-26-03682-f003:**
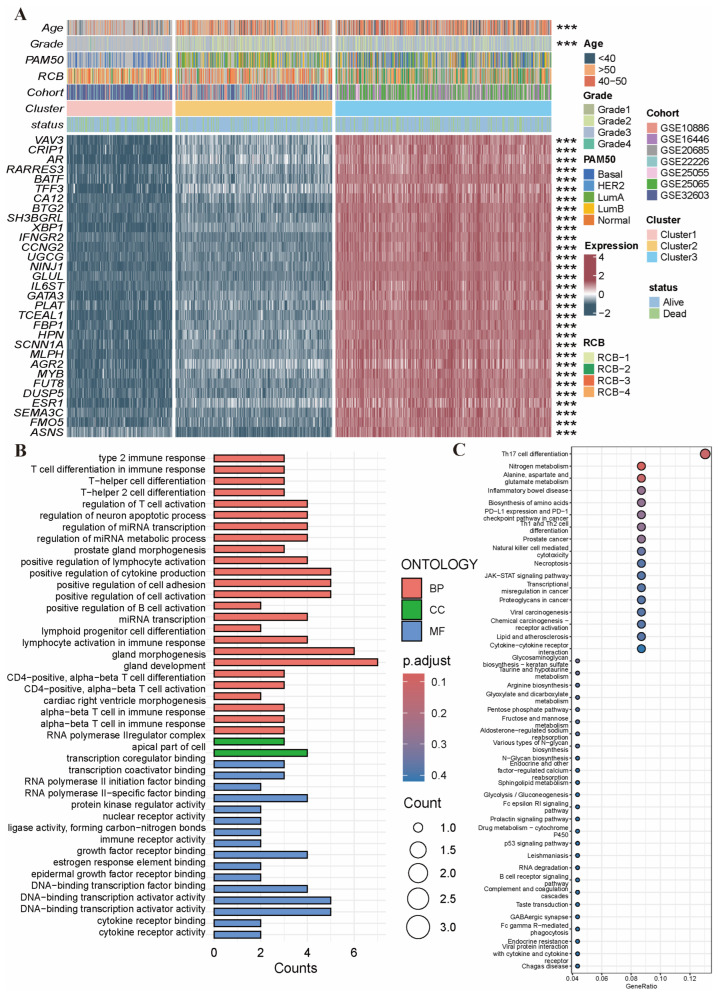
Variant landscape of PCD-related DEGs in BRCA patients. (**A**) Differences in clinical characteristics as well as the expression levels of PCD-related DEGs among 3 PCD clusters. ***, *p* < 0.001. (**B**) GO enrichment analysis based on PCD-related DEGs. (**C**) KEGG enrichment analysis based on PCD-related DEGs. Abbreviations: PCD-related DEGs: programmed cell death-related differentially expressed genes; BRCA: breast cancer; GO, gene ontology; KEGG, kyoto encyclopedia of genes and genomes.

**Figure 4 ijms-26-03682-f004:**
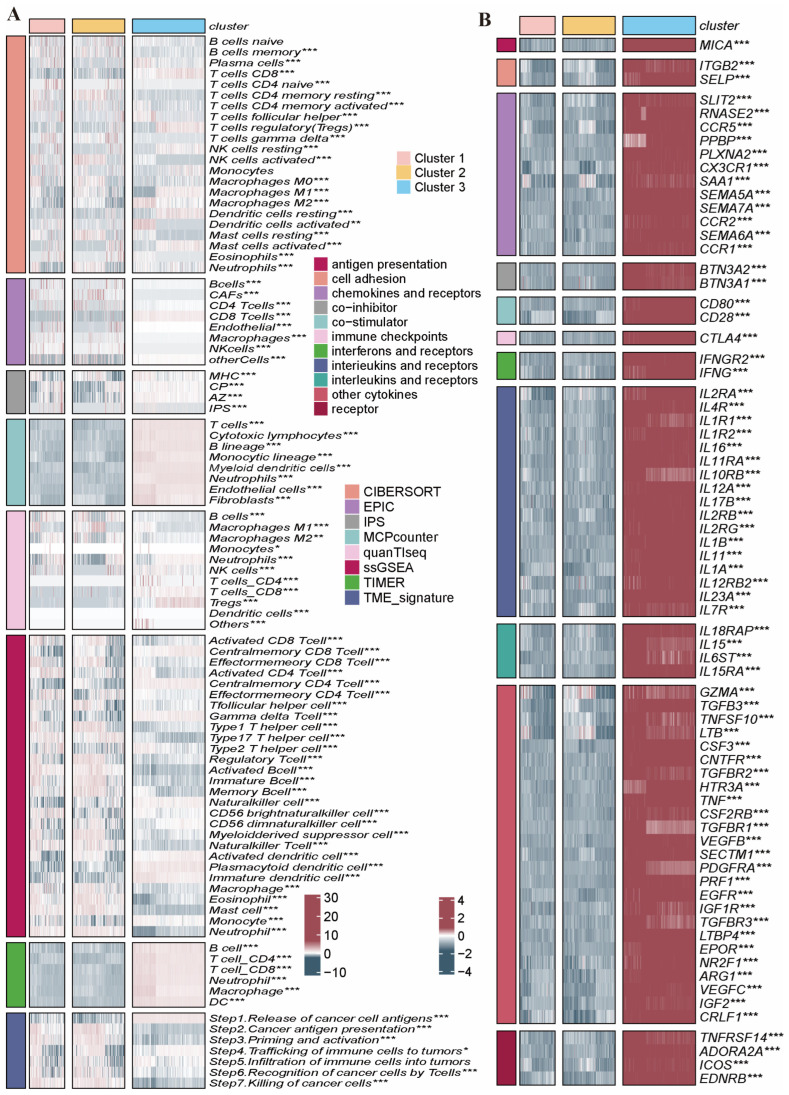
Immune infiltration characteristics in the distinct PRG clusters. (**A**) Differences in immune cell infiltration among PRG clusters with different immune infiltration analysis tools. (**B**) Differential expressions of multiple immune-related cytokines and immune checkpoints in PRG clusters. *, *p* < 0.05; ** *p* < 0.01, ***, *p* < 0.001. Abbreviations: PRGs, programmed cell death-related genes.

**Figure 5 ijms-26-03682-f005:**
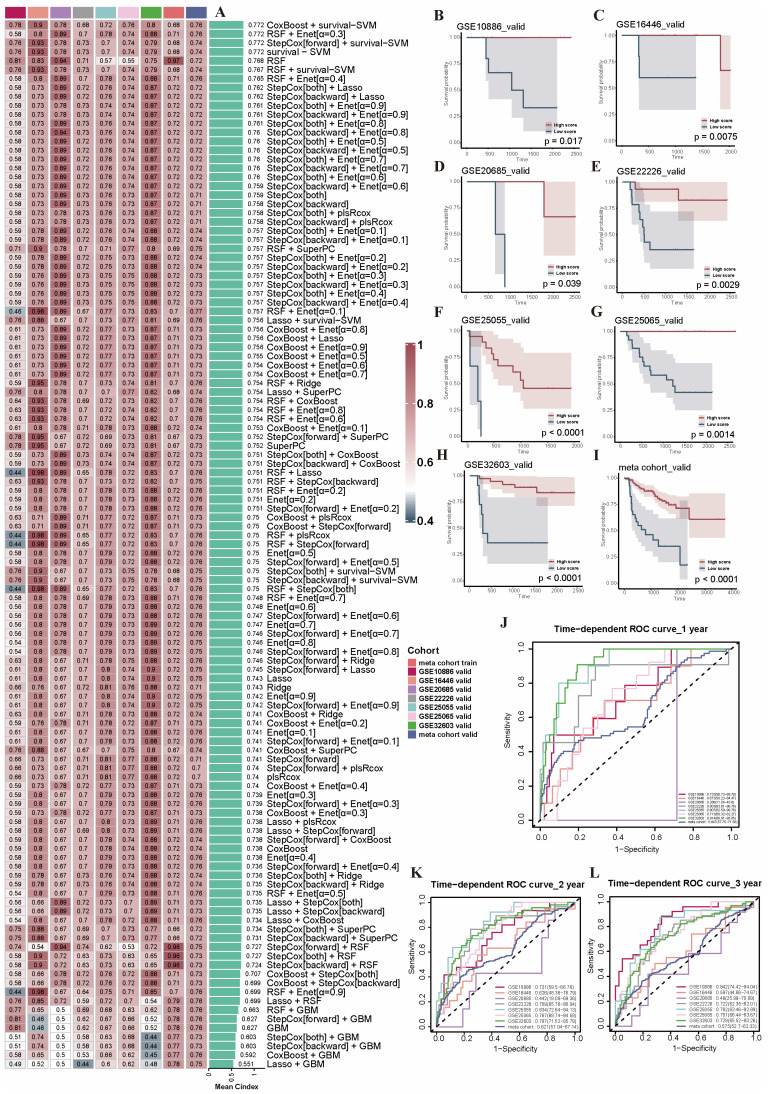
The construction and evaluation of the PRGs prognostic risk model. (**A**) In total, 117 combinations of machine learning algorithms. Sorted by the mean C-index. (**B**–**I**) Kaplan–Meier survival curve of 7 independent and meta cohort validation sets. (**J**–**L**) Time–ROC curves for 1 to 3 years in the 7 independent and meta cohort validation sets. Abbreviations: PRGs, programmed cell death-related genes; Time-ROC, time-dependent receiver operating characteristic.

**Figure 6 ijms-26-03682-f006:**
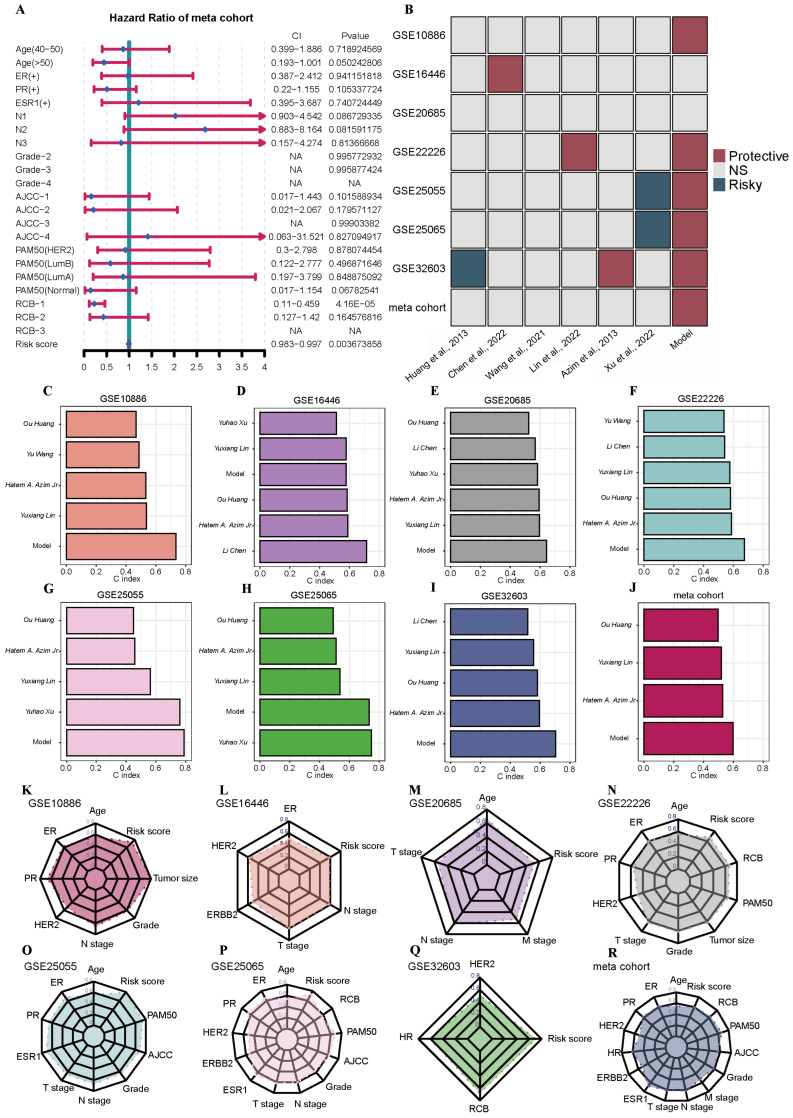
Multivariate Cox regression analysis of the model and comparison with other models. (**A**) Multivariate Cox regression validates the value of the model risk score as an independent prognostic indicator for BRCA patients. (**B**) Survival correlation analyses of different models in 7 independent and meta cohort datasets. (**C**–**J**) A comparison of the C-index of the models in 7 independent and meta cohort datasets. (**K**–**R**) A comparison of the C-index between model risk score and clinical factors in 7 independent and meta cohort datasets. The other six models in B were drawn from the following six references: [18,19,20,21,22,23]. Abbreviations: BRCA: breast cancer.

**Figure 7 ijms-26-03682-f007:**
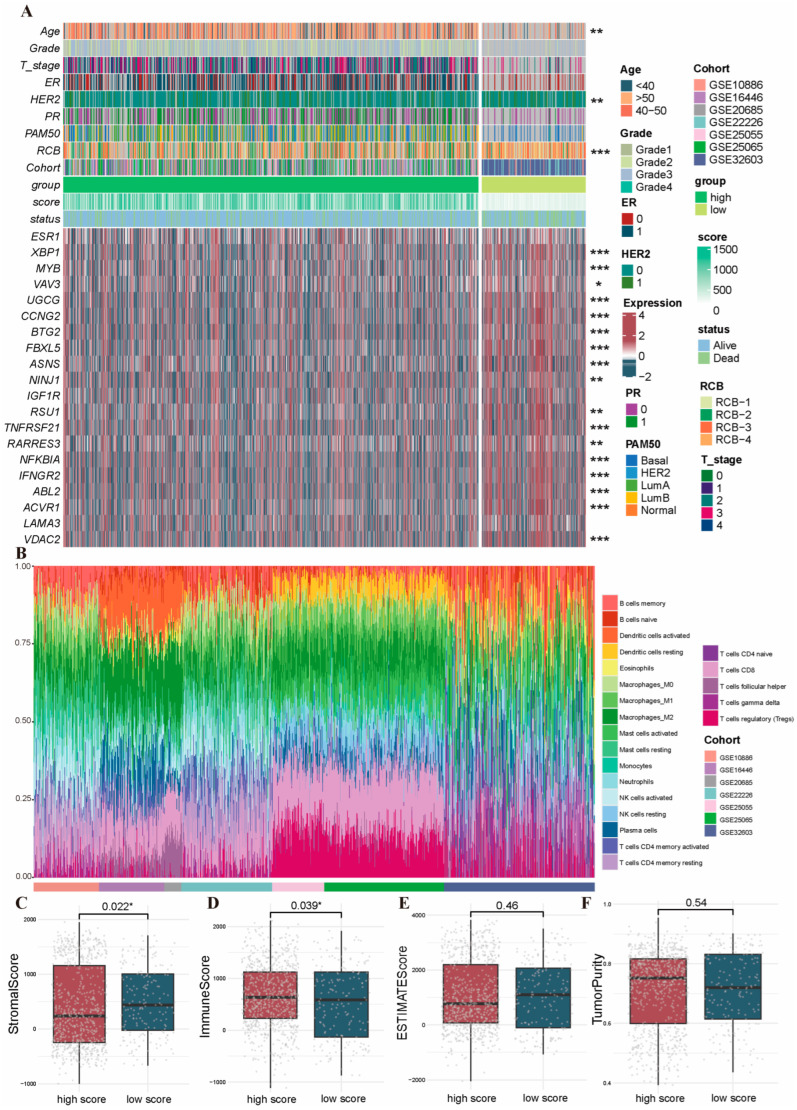
Clinicopathologic features and tumor immune microenvironment analysis based on the PRGs prognostic risk model. (**A**) The heatmap demonstrating differences in 20 model gene expression levels and multiple clinical factors between high- and low-risk subgroups. (**B**) Proportional stacking plots of immune cell components in the different datasets. (**C**–**F**) Comparisons between the 2 subgroups in terms of (**C**) stromal score, (**D**) immune score, (**E**) ESTIMATE score, and (**F**) tumor purity. *, *p* < 0.05; **, *p* < 0.01; ***, *p* < 0.001. Abbreviations: PRGs, programmed cell death-related genes.

**Figure 8 ijms-26-03682-f008:**
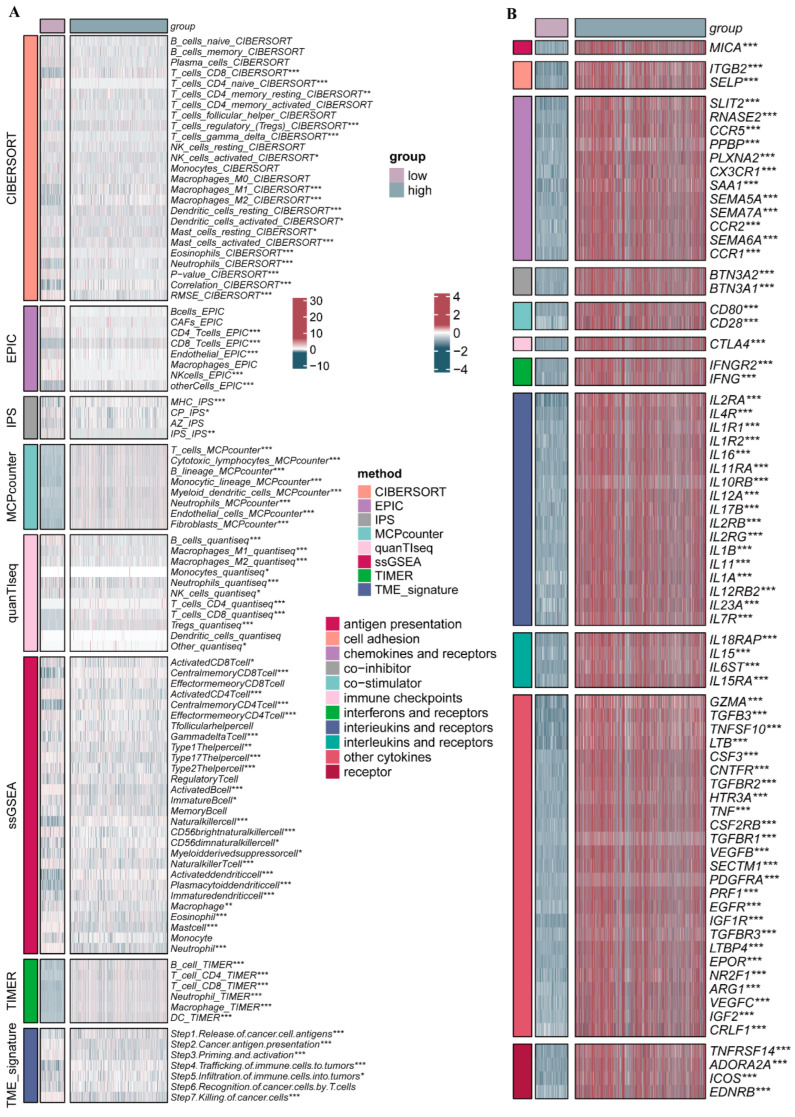
Analysis of immune infiltration characteristics between high- and low-risk subgroups. (**A**) Differences in immune cell infiltration between two subgroups with different immune infiltration analysis tools. (**B**) Differential expressions of multiple immune-related cytokines and receptors between two subgroups. * *p* < 0.05, ** *p* < 0.01, ***, *p* < 0.001.

**Figure 9 ijms-26-03682-f009:**
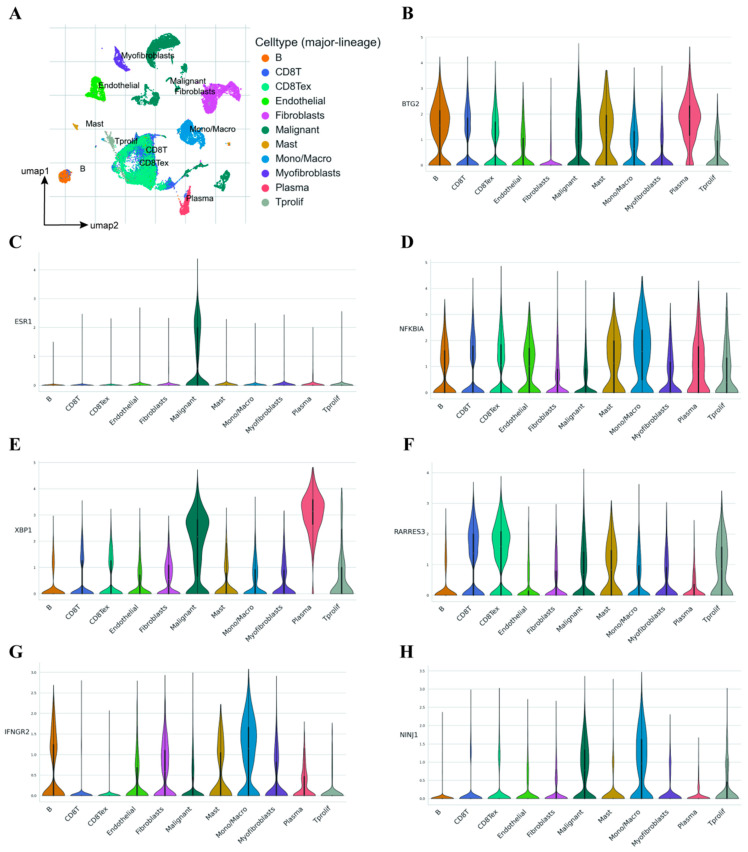
Single-cell transcriptome validation. (**A**) UMAP for dimensionality reduction in data derived from the EMTAB8107 dataset. (**B**–**H**) The distribution of model genes among different cell populations. Abbreviations: UMAP, uniform manifold approximation and projection.

**Figure 10 ijms-26-03682-f010:**
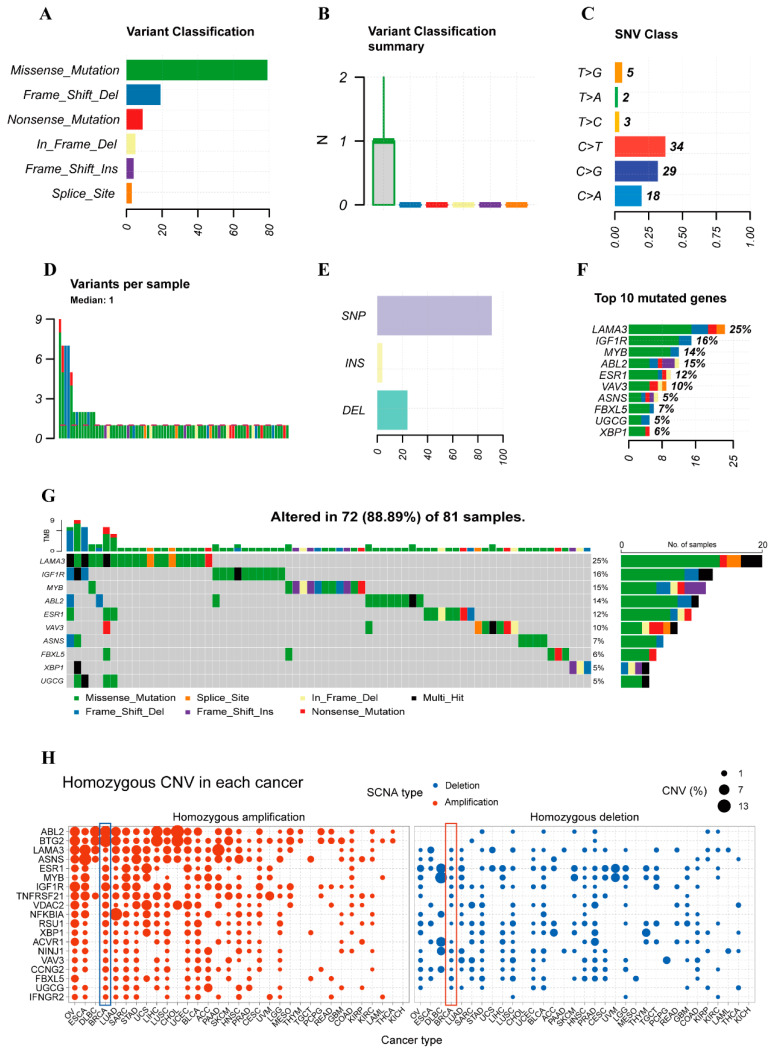
Mutation analysis of model genes. (**A**,**B**) The ratio and quantity of SNP. (**C**) Rates of different cases of SNP methyl transformations. (**D**) Mutation status of each sample. (**E**) The proportion of mutations accounted for by single nucleotide alterations, insertions, and deletions. (**F**,**G**) The top 10 mutated genes in BRCA. (**H**) CNV mutation frequency of model genes in 31 types of cancers. Breast cancer is circled in the blue and red box. Abbreviations: SNP, single nucleotide polymorphism; BRCA: breast cancer; CNV, copy number variation.

**Figure 11 ijms-26-03682-f011:**
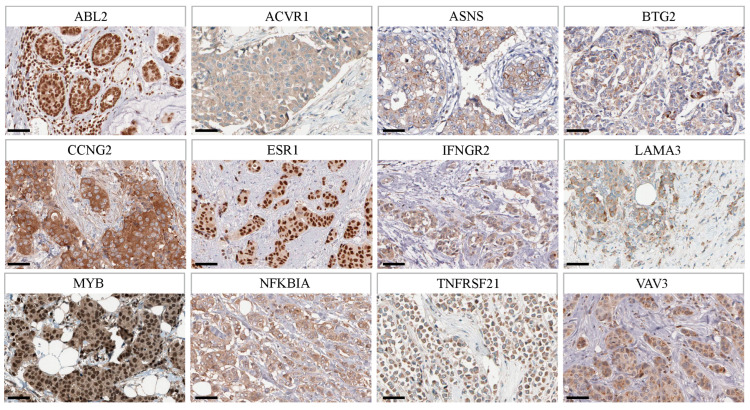
Validation of immunohistochemical staining from HPA database. Scale bar = 50 μm. Abbreviations: HPA, human protein atlas.

**Figure 12 ijms-26-03682-f012:**
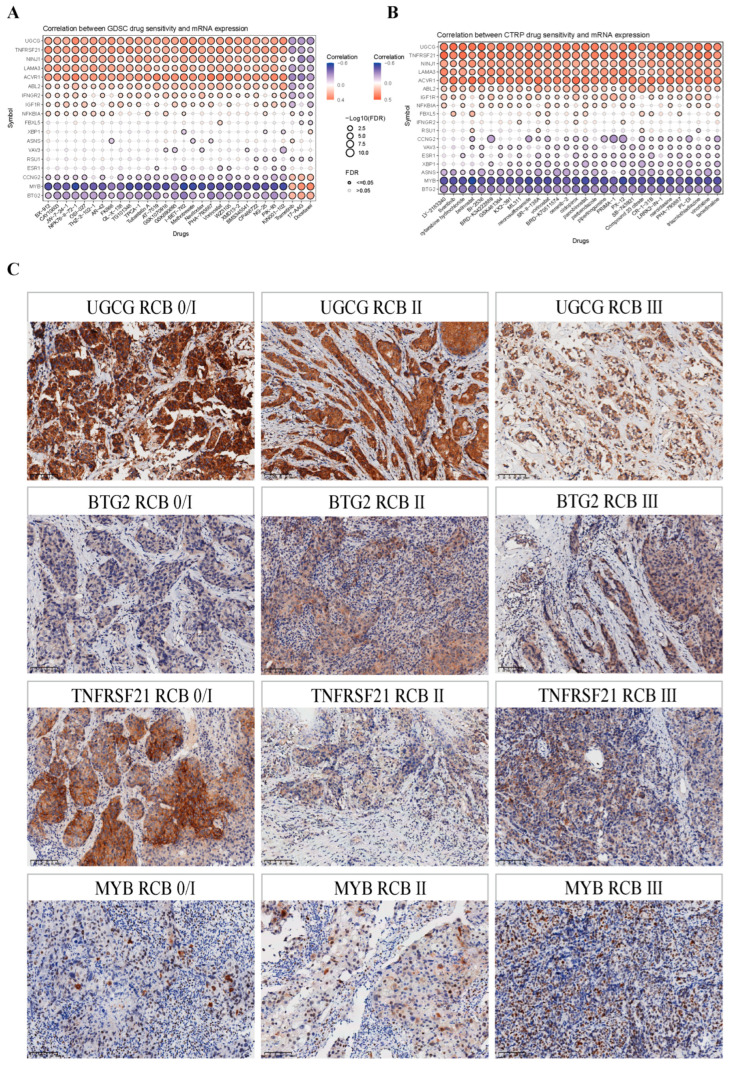
The prognostic model is associated with the response to NAC in BRCA. (**A**,**B**) Pharmacological analysis. Bubble plot of the relationship between drugs and model genes in the GDSC database (**A**) and CTRP database (**B**). (**C**) IHC staining of UGCG, TNFRSF21, MYB, and BTG2 in biopsy tissues of patients with different RCB grades before NAC. Scale bar = 100 μm. Abbreviations: NAC, neoadjuvant chemotherapy; BRCA: breast cancer; GDSC, genomics of drug sensitivity in cancer; CTRP, cancer therapeutics response portal; IHC, immunohistochemistry; RCB, residual cancer burden.

**Table 1 ijms-26-03682-t001:** Relationship between RCB classification and clinicopathological features of BRCA before NAC.

Characteristic	Cases (*n* = 56)	RCB-0 + RCB-I (*n* = 31)	RCB-II (*n* =12)	RCB-III (*n* = 13)	*p*
Age (years)					0.330 a
≤50	31	18	8	5	
>50	25	13	4	8	
ER					0.525 b
Positive	6	2	2	2	
Negative	50	29	10	11	
PR					0.642 b
Positive	7	3	2	2	
Negative	49	28	10	11	
HER2					0.375 a
Positive	32	18	5	9	
Negative	24	13	7	4	
UGCG					0.019 b *
Low	10	2	3	5	
High	46	29	9	8	
BTG2					0.012 a *
Low	33	23	3	7	
High	23	8	9	6	
TNFRSF21					<0.001 b *
Negative	20	7	10	3	
Positive	36	24	2	10	
MYB					0.029 a *
Negative	33	23	4	6	
Positive	23	8	8	7	
Ki-67(%)					0.244 b
≤20	3	1	0	2	
>20	53	30	12	11	
Molecular subtype					0.524 b
HR+ and HER2–	2	2	0	0	
HER2+	32	18	5	9	
Triple-negative	22	11	7	4	
Lymph nodes metastasis					<0.001 a *
No metastasis	28	20	8	0	
Metastasis	28	11	4	13	

RCB, Residual Cancer Burden; ER, Estrogen Receptor; PR, Progesterone Receptor; HER2, Human Epidermal Growth Factor Receptor 2; UGCG, UDP-Glucose Ceramide Glucosyltransferase; BTG2, B-cell Translocation Gene 2; TNFRSF21, Tumor Necrosis Factor Receptor Superfamily Member 21; MYB, MYB Proto-Oncogene. a, chi-square test was used for analysis; b, Fisher’s exact test was used for analysis; *, *p* < 0.05.

**Table 2 ijms-26-03682-t002:** Correlation analysis between NAC efficacy and pre-NAC clinicopathological features of BRCA.

Characteristic	Age	ER	PR	HER2	UGCG	BTG2	TNFRSF21	MYB	Ki-67	Lymph Nodes Metastasis
r	−0.061	−0.153	−0.095	0.021	0.332	−0.346	0.305	−0.346	0.105	−0.323
*p*	0.657	0.259	0.486	0.879	0.013 *	0.009 *	0.022 *	0.009 *	0.439	0.015 *

ER, Estrogen Receptor; PR, Progesterone Receptor; HER2, Human Epidermal Growth Factor Receptor 2; UGCG, UDP-Glucose Ceramide Glucosyltransferase; BTG2, B-cell Translocation Gene 2; TNFRSF21, Tumor Necrosis Factor Receptor Superfamily Member 21; MYB, MYB Proto-Oncogene; *, *p* < 0.05.

## Data Availability

The datasets used in this study can be found online as described above.

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
