# Peer review of "Integrating Machine Learning and Bulk and Single-Cell RNA Sequencing to Decipher Diverse Cell Death Patterns for Predicting the Prognosis of Neoadjuvant Chemotherapy in Breast Cancer"

_ijms, 2025, doi:10.3390/ijms26083682_

Round 1

Reviewer 1 Report

Comments and Suggestions for Authors

The study by Xiang and Yang et al. applied machine learning on integrated bioinformatics data to predict therapeutical responses in breast cancer patients regarding programmed death patterns and related gene expression alterations.

This is a fascinating study, in my opinion. Worth noting is the high-quality processing and integration of data derived from various databases and technologies, which is quite rare practice. Perhaps authors could mention that fact less often in their manuscript.

I would suggest some modifications to the discussion as I lack a more functional/biological point of view on the findings, which could reflect the type of predicted response. Gene symbols should be in italics.

Author Response

Comments 1: Worth noting is the high-quality processing and integration of data derived from various databases and technologies, which is quite rare practice. Perhaps authors could mention that fact less often in their manuscript.

Response 1: Thank you for pointing this out. We agree with this comment. Therefore we have deleted the unused database. We deleted the original content on page 3, line 100 of the manuscript: the Genotype-Tissue Expression (GTEx) (https://www.gtexportal.org/home/), and the European Molecular Biology Laboratory (EMBL) (https://www.ebi.ac.uk/biostudies/), among others. We have kept the revision traces in the revised manuscript for your better review.

Comments 2: I would suggest some modifications to the discussion as I lack a more functional/biological point of view on the findings, which could reflect the type of predicted response.

Response 2: Thank you for pointing this out. We agree with this comment. Therefore we have added relevant mechanistic descriptions in the discussion section. We added the following on page 27, line 572 of the manuscript: Some of these genes have been reported to be potentially associated with NAC efficacy in BRCA. For example, tumors with high ESR1 expression often have a lower pCR rate, possibly because such tumors rely on hormonal signaling rather than chemotherapy-sensitive pathways[43]. In addition, the IGF1R rs2016347 T allele and chemotherapy-induced IGF1R downregulation were jointly associated with improved NAC response[44]. However, the bond between some other PCD-related DEGs and NAC in BRCA is not yet understood. However, the relationship between most PCD-related DEGs and NAC and the mechanism of action were not found in the literature. This is also the innovation of our article and inspires us to carry out more studies on the mechanisms by which other genes affect NAC efficacy. We have also added the following on page 28, line 648 of the manuscript: UGCG’s negative correlation with RCB may stem from GlcCer accumulation, which is the precursor of all glycosphingolipids (GSL) complexes. GSL is not only an important membrane component, but also participates in apoptosis as a signaling molecule[55]. TNFRSF21 internalization in acidic microenvironment recruits caspase-8/GSDMC, inducing pyroptosis and correlating with lower RCB[56]. We have kept the revision traces in the revised manuscript for your better review.

Comments 3: Gene symbols should be in italics.

Response 3: Thank you for pointing this out. We agree with this comment. We have italicized all gene symbols in the revised manuscript. The corresponding protein symbols were not italicized. We have kept the revision traces in the revised manuscript for your better review.

Reviewer 2 Report

Comments and Suggestions for Authors

The article by Xiang et al. aims to integrate machine learning, bulk, and single-cell RNA sequencing to develop a prognostic model for predicting the outcome of neoadjuvant therapy in breast cancer patients. The study is highly interesting and employs innovative approaches such as machine learning, making it suitable for the journal. I have a few comments:

  • The introduction is quite long and somewhat unfocused. I suggest streamlining it to make it more concise, ensuring the rationale remains clear without overly detailed explanations of cell death mechanisms. If the authors believe this level of detail is essential, they could include it as supplementary material. As it stands, the focus on the study’s rationale risks being diluted, giving the impression of a review rather than a research article.
  • For the graphical abstract and other figures, I recommend adding explanations for the abbreviations used in the captions to enhance clarity.
  • The manuscript would benefit from careful proofreading, as there are some typographical errors, such as "survial" in the graphical abstract.
  • Gene names should be italicized throughout the text.
  • The authors state that these findings require validation through randomized trials, larger cohorts, and longer follow-ups. I would like to ask the authors how they envision the translational potential of this approach — specifically, how they see this model being implemented in clinical practice.

Author Response

Comments 1: The introduction is quite long and somewhat unfocused. I suggest streamlining it to make it more concise, ensuring the rationale remains clear without overly detailed explanations of cell death mechanisms. If the authors believe this level of detail is essential, they could include it as supplementary material. As it stands, the focus on the study’s rationale risks being diluted, giving the impression of a review rather than a research article

Response 1: Thank you for pointing this out. We agree with this comment. Therefore we have removed the description of 19 PCD patterns in the introduction. We deleted the original content on page 2, line 66 of the manuscript: Many of the PCD pathways involve caspase activation. For example, triggered through the activation of death receptors or the presence of stress signals. This triggers the activation of caspase-8 and pro-apoptotic proteins, which in turn cause mitochondrial outer membrane permeabilization. Ultimately, this cascade of events results in cell death[7]. Similarly, activation of inflammasomes such as AIM2 and the NLR family leads to the recruitment of downstream caspase-1, which in turn leads to tumor cell lysis. This is the mechanism by which pyroptosis induces the death of tumor cells[8]. Anoikis is triggered when integrin-mediated cell attachment is lost[9]. This activates either the intrinsic pathway, involving mitochondrial disruption, or the external signaling pathway, which is regulated by death receptors. Both pathways ultimately lead to the activation of caspases. In some cases, anoikis can also occur through caspase-independent mechanisms, involving the release of Bit-1 from the mitochondria[9]. In addition, there are some patterns of death that are combinations of certain PCD pathways. For instance, PANoptosis encompasses the concurrent triggering of pyroptosis, apoptosis, and necroptosis. This mechanism is regulated by the PANoptosome complex, which incorporates essential elements from each of these pathways[10]. Besides, there is also a subset of PCD pathways that do not depend on casepases. To illustrate, necroptosis can be activated by RIPK1 and RIPK3, resulting in the phosphorylation of downstream MLKL to form multimers, which in turn leads to tumor cell death[11]. Ferroptosis is driven by membrane lipid peroxidation leading to plasma membrane rupture, regulated by the GSH-GPX4 and NADPH-FSP1 systems[12, 13]. Another PCD related to metal ions is cuproptosis. This process is triggered by an excess of copper binding to lipoylated tricarboxylic acid cycle proteins in the mitochondria. As a result, protein oligomerization occurs, leading to the destabilization of Fe-S cluster proteins. This causes proteotoxic stress, which ultimately results in cell death[14]. In addition, autophagy is initiated by AMPK-mediated inhibition of mTORC1, leading to LC3-II insertion into autophagosomes and their merging with lysosomes to create autolysosomes, enabling cellular breakdown[15]. Parthanatos is initiated by the overactivation of PARP-1 in response to DNA damage, producing PAR that signals mitochondrial release of AIF, which migrates to the nucleus and causes extensive DNA fragmentation[16]. Lysosome-dependent cell death is initiated by the permeabilization of the lysosomal membrane. This event causes the release of cathepsins and other hydrolases into the cytosol. Once in the cytosol, these enzymes degrade cellular components, ultimately leading to cell death[17].Alkaliptosis is driven by intracellular alkalinization and is regulated by the IKBKB-NF-κB-dependent carbonic anhydrase 9 (CA9) pathway, leading to pH-dependent cell death in cancer cells[18]. Oxeiptosis is induced by reactive oxygen species (ROS) and regulated through the KEAP1/PGAM5/AIFM1 pathway, leading to caspase-independent cell death[19]. NETosis, marked by the expulsion of neutrophil extracellular traps (NETs), web-like formations released in response to infection or damage, can be triggered by amiloride derivatives like EIPA and MIA[20, 21]. The cell death caused by this process is often referred to as netotic cell death. Immunogenic cell death (ICD) is characterized by the release of DAMPs like calreticulin, ATP, and HMGB1, which activate dendritic cells and T-cell responses, ultimately leading to tumor cell death[22]. Paraptosis is known for dysfunction in both the endoplasmic reticulum (ER) and mitochondria. ER stress arises from the accumulation of misfolded proteins, while mitochondrial dysfunction is caused by Ca2+ overload and increased ROS. These interconnected processes can occur simultaneously, ultimately leading to cell death[23]. Methuosis involves the accumulation of vacuoles from macropinocytic activity due to dysfunctional endosomal trafficking, resulting in the creation of large vacuoles that fill the cytoplasm and cause caspase-independent cell death[24]. Entosis refers to the engulfing of neighboring cells by one cell through actomyosin contraction and Rho-ROCK signaling. During this process, the invading cells are internalized by the host cells, and most internalized cells undergo entotic cell death[25, 26]. We have kept the revision traces in the revised manuscript for your better review.

Comments 2: For the graphical abstract and other figures, I recommend adding explanations for the abbreviations used in the captions to enhance clarity.

Response 2: Thank you for pointing this out. We agree with this comment. Therefore we have annotated the full name of the corresponding abbreviation below each image. We have kept the revision traces in the revised manuscript for your better review.

Comments 3: The manuscript would benefit from careful proofreading, as there are some typographical errors, such as "survial" in the graphical abstract.

Response 3: We were really sorry for our careless mistakes. Thank you for your reminder. We have changed "survial" to "survival" in Figure 1 and carefully proofread the manuscript. We have kept the revision traces in the revised manuscript for your better review.

Comments 4: Gene names should be italicized throughout the text.

Response 4: Thank you for pointing this out. We agree with this comment. We have italicized all gene symbols in the revised manuscript. The corresponding protein symbols were not italicized. We have kept the revision traces in the revised manuscript for your better review.

Comments 5: The authors state that these findings require validation through randomized trials, larger cohorts, and longer follow-ups. I would like to ask the authors how they envision the translational potential of this approach — specifically, how they see this model being implemented in clinical practice.

Response 5: Thank you for pointing this out. We agree with this comment. Therefore we have added the following on page 28, line 659 of the manuscript: Our PRGs-based prognostic model holds significant translational potential in clinical settings, particularly in guiding NAC treatment strategies for BRCA patients. Through the use of  this model,  clinicians could better stratify patients undergoing NAC, predict treatment response, and optimize therapeutic decisions accordingly. Patients identified as high-risk based on the PRG score may benefit from more intensive or alternative therapeutic regimens, such as incorporating immune checkpoint inhibitors or targeted therapies. Conversely, low-risk patients with a favorable PRG profile might achieve sufficient tumor regression with standard NAC regimens, potentially avoiding overtreatment and its associated toxicities. This stratification would facilitate a more tailored approach to patient management, improving outcomes while minimizing unnecessary treatment burdens. By combining this model with conventional clinical and pathological features, such as hormone receptor status, HER2 expression, and Ki-67 index, clinicians could make more informed decisions regarding NAC suitability and post-treatment management. Furthermore, as single-cell transcriptomics advances, using insights from scRNA-seq into the model could refine its predictive accuracy, helping to identify patients who may benefit from novel therapeutic combinations. We have also added the following on page 29, line 681 of the manuscript: The detection of PCD-related DEGs in this model is feasible using standard molecular techniques such as scRNA-seq, quantitative PCR (qPCR), or IHC. To enhance the clinical impact of this model, further prospective validation in larger, multi-center cohorts with long-term follow-up is necessary.We have kept the revision traces in the revised manuscript for your better review.

Round 2

Reviewer 2 Report

Comments and Suggestions for Authors

Manuscript can be accepted in current form.